# ATTENTION SINKS AS INTERNAL SIGNALS FOR HALLUCINATION DETECTION IN LARGE LANGUAGE MODELS

## ABSTRACT

Large language models frequently exhibit hallucinations: fluent and confident outputs that are factually incorrect or unsupported by the input context. While recent hallucination detection methods have explored various features derived from attention maps, the underlying mechanisms they exploit remain poorly understood. In this work, we propose SinkProbe, a hallucination detection method grounded in the observation that hallucinations are deeply entangled with attention sinks—tokens that accumulate disproportionate attention mass during generation—indicating a transition from distributed, input-grounded attention to compressed, prior-dominated computation. Importantly, although sink scores are computed solely from attention maps, we find that the classifier preferentially relies on sinks whose associated value vectors have large norms. Moreover, we show that previous methods implicitly depend on attention sinks by establishing their mathematical relationship to sink scores. Our findings yield a novel hallucination detection method grounded in theory that produces state-of-the-art results across popular datasets and LLMs.

## 1 INTRODUCTION

Large language models (LLMs) have achieved remarkable success across a wide range of natural language understanding and generation tasks, and are increasingly deployed in settings that require factual reliability, such as question answering, summarization, and decision support (22; 24; 42). Despite these advances, LLMs remain prone to *hallucinations*—fluent and confident outputs that are factually incorrect, unverifiable, or unsupported by the input context (28; 14).

Hallucinations pose a fundamental challenge to the safe and trustworthy use of LLMs. Unlike traditional task-specific models, modern LLMs operate in open-ended regimes, where errors often manifest as plausible fabrications rather than explicit contradictions (20). A number of existing approaches address hallucinations at the *output level*. These include fact-checking against external knowledge sources, retrieval-augmented generation, consistency checks across multiple sampled responses, and uncertainty-based metrics derived from token probabilities or entropy (32; 26; 14; 13; 35). While effective in certain settings, such methods typically require additional resources such as external corpora, or multiple generations, and provide limited insight into the internal computational mechanisms that give rise to hallucinations.

Recently, a complementary line of work has explored *internal signals* of hallucination by analyzing attention maps, hidden representations, or spectral properties of Transformer-Decoder models (38). These methods exploit the observation that hallucinated generations are often associated with atypical internal representations (9; 12), lack of grounding of generations (10; 7), or invalid attention dynamics (36; 8; 15). In this work, we trace hallucinations to breakdowns in internal information flow captured by attention sink scores, and show that this perspective both unifies prior detection methods and yields a simple and effective hallucination detector.

*Attention sinks* (40; 17)—tokens attracting disproportionate attention despite low semantic relevance—are a pervasive mechanism for compression and information routing in Transformers (5; 33). We hypothesize that hallucinations may stem from breakdowns in this internal flow, rather than knowledge gaps alone. Consequently, we introduce the *attention sink score*: an interpretable diagnostic derived from cross-layer attention statistics. This yields a compact, token-agnostic signal for detecting hallucinations.

Figure 1: Pipeline for hallucination detection based on attention sink scores. For each layer $l$ and head $h$, we compute sink scores $\mathbf{s}^{(l,h)}$, defined as the average of attention scores directed toward each token position. These scores are then sorted, and the top-$k$ values are selected as features. The selected features from all layers and heads are concatenated to form the final feature vector $\mathbf{z}$, which is passed to a hallucination probe – a binary classifier trained on labeled examples to distinguish hallucinated outputs from factually correct ones at inference time.

Importantly, we show that not all tokens with large attention are equally relevant for hallucination detection. While sink scores are computed solely from attention maps, their computational impact depends on the associated value vectors. We find that the hallucination signal is concentrated in tokens where high attention concentration coincides with unusually large value norms—a subset of computationally active tokens that dominate the attention output and induce compressed, prior-dominated representations.

We further provide a unifying perspective on existing attention-based hallucination detectors, demonstrating that several spectral and graph-based methods can be interpreted as transformations of sink behavior. Empirically, we show that sink-score–based features achieve superior hallucination detection performance across multiple models and benchmarks, consistently outperforming state-of-the-art hallucination detectors.

In summary, our main contributions are as follows:

- We propose SinkProbe - a novel method for hallucination detection from attention maps
- We identify computationally active sinks, showing that sink-based signals are strongest when attention concentration coincides with large value vector norms.
- We show that sink-score–based features achieve state-of-the-art hallucination detection across several common LLMs and benchmarks.

## 2 METHOD

We leverage the attention sink score, an attention-based measure derived from standard attention maps that quantifies the concentration of attention on the tokens. Our method operates purely on internal attention weights and produces a compact feature representation suitable for hallucination detection. We call this method SinkProbe and introduce it in the following section by first formalizing sink scores at the token, head, and layer levels, then describing how they are aggregated into a probe feature vector. The method is summarized in Figure 1.

**Attention Sinks.** A pivotal component of our methodology is the *attention sink score*. We leverage this measure—primarily developed to detect attention sinks (40; 17), i.e., tokens that accumulate significant attention from future contexts while contributing minimal semantic value—to trace the information flow within LLMs. Let $\mathbf{A}^{(l,h)} \in \mathbb{R}^{T \times T}$ denote the attention matrix produced by attention head $h \in \{1, \ldots, H\}$ in layer $l \in \{1, \ldots, L\}$, where $A_{u,i}^{(l,h)}$ represents the attention weight from token $u$ to token $i$, where $u, i \in \{1, \ldots, T\}$, and $T$ is the sequence length. Matrix $A^{(l,h)}$ is row-stochastic and strictly causal, i.e., $A_{u,i}^{(l,h)} = 0$ for all $i > u$. To quantify sink behavior, we use *sink score* for each token, as defined by (17).

**Definition 2.1** (Sink Score, 17). For a given attention head $(l, h)$, the sink score of token $i$ is defined,

$$s_i^{(l,h)} = \frac{1}{T-i} \sum_{u=i}^{T} A_{u,i}^{(l,h)}. \tag{1}$$

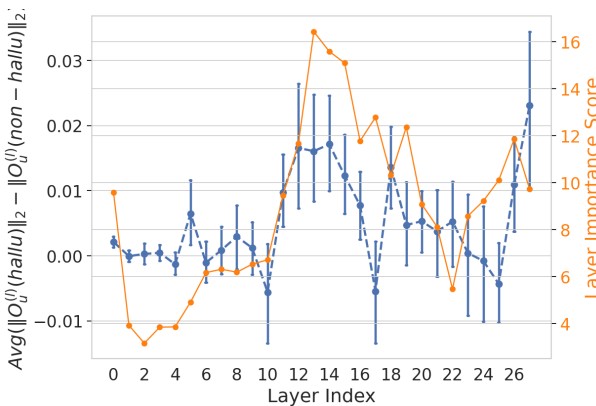

Figure 2: Relationship between attention output norm differences and layer importance. The blue dashed curve shows the average difference in attention output norms $\|O_u^{(l)}\|_2$ between hallucinated and non-hallucinated examples across layers, with error bars indicating standard error. The orange solid curve reports layer importance scores derived from the hallucination probe. Both signals peak in the middle layers, indicating strong alignment between attention-level anomalies and probe-identified layer importance.

This quantity measures the average amount of attention token $i$ receives from all subsequent tokens, normalized by the remaining sequence length. High sink scores indicate tokens that persistently attract attention as generation progresses. Sink scores are defined independently for each layer and head, yielding a tensor $S \in \mathbb{R}^{L \times H \times T}$.

**From Sink Scores to Features.** Sink scores typically exhibit a highly skewed distribution: in most attention heads, only a small number of tokens accumulate large scores, while the majority receive negligible attention. Rather than relying on token identity or position, we summarize this structure using order statistics. For each head $(l, h)$, we sort the sink scores $\tilde{s}^{(l,h)} = \text{sort}\big(s^{(l,h)}\big)$, and retain the top-$k$ values. The final feature vector for a given example is constructed by concatenating these values across all layers and heads:

$$z = \bigoplus_{l=1}^{L} \bigoplus_{h=1}^{H} \big[\tilde{s}_T^{(l,h)}, \tilde{s}_{T-1}^{(l,h)}, \ldots, \tilde{s}_{T-k+1}^{(l,h)}\big] \in \mathbb{R}^{L \cdot H \cdot k}. \tag{2}$$

This representation captures the degree of attention concentration while remaining agnostic to token semantics, sequence length, and model architecture. To evaluate the predictive power of sink scores, we train a lightweight hallucination probe, i.e., logistic regression classifier.

## 2.1 ROLE OF VALUE VECTORS IN ATTENTION SINK BEHAVIOR

While attention sink scores characterize the structural concentration of attention weights in Transformer Decoder, their effect on the model's computation depends critically on the associated value vectors. For a given attention head $(l, h)$, the output at token position $u$ is given by

$$O_u^{(l,h)} = \sum_{i=1}^{T} A_{u,i}^{(l,h)} V_i^{(l,h)}$$

A token $i$ acting as an attention sink contributes to this output in proportion to both its incoming attention mass and the magnitude of its value vector. Attention sinks, which serve to prevent over-mixing (5) and are often associated with semantically vacuous tokens (e.g., the `<bos>` token), have been shown to exhibit small value-vector magnitudes (17). Consequently, a sink with small $\|V_i^{(l,h)}\|$ exerts limited influence on $O_u^{(l,h)}$, even when it attracts substantial attention, whereas a sink with a large value norm can dominate the attention output across multiple positions. This distinction implies that only a subset of attention sinks are *computationally active*—that is, they significantly influence hidden representations through repeated injection of high-magnitude value vectors.

Motivated by this observation, we examined the norms of attention outputs $\|O_u^{(l,h)}\|_2$. Specifically, we traced these output vectors and computed their norms for tokens identified as important by our hallucination probe (see Section D.1), partitioning the data according to whether each example was hallucinated or non-hallucinated. Section 2 presents the aggregated norm differences alongside probe importance scores for two selected datasets, stratified by layer. The observed pattern suggests that sink-score–based features primarily capture sinks whose attention concentration coincides with elevated value magnitudes, rendering them more likely to exert substantial influence on the

attention output. Although this association is not universal—as evidenced by divergent behavior in certain layers—it provides a principled account of why only a subset of attention sinks consistently contributes to hallucination detection, despite the widespread presence of sink behavior across heads and layers.

## 3 EXPERIMENTS

**Experimental set-up.** We evaluate our method on seven diverse hallucination detection benchmarks spanning question answering and mathematical reasoning. We evaluate our method on four widely used open-weight LLMs spanning three model families and ranging from 3B to 12B parameters: Llama-3.2-3B (2), Phi-3.5 (4B) (1), Llama-3.1-8B (2), and Mistral-Nemo (12B) (27). We compare SinkProbe against six state-of-the-art attention-based hallucination detection baselines: (1) AttentionScore (36), (2) AttnLogDet, (3) AttnEigvals and (4) LapEigval (8), (5) LookbackLens (10), (6) MTopDiv (7), All baselines except the unsupervised AttentionScore use the same logistic regression probe architecture and training protocol to ensure fair comparison. Further details on the experimental set-up are provided in Appendix.

Table 1: Results of hallucination detection experiments, the values represent mean and standard deviation of ROC-AUC scores over 5-fold cross-validation, Δ represents relative improvement of SinkProbe compared with second best or best performing method.

| LLM | Dataset Feature | GSM8K | HaluevalQA | NQOpen | SQuADv2 | TriviaQA | TruthfulQA | UMWP |
|---|---|---|---|---|---|---|---|---|
| Phi3.5 | AttnScore | $0.741 \pm 0.069$ | $0.712 \pm 0.009$ | $0.729 \pm 0.022$ | $0.678 \pm 0.015$ | $0.705 \pm 0.016$ | $0.693 \pm 0.045$ | $0.757 \pm 0.022$ |
| | AttnLogDet | $0.782 \pm 0.058$ | $0.805 \pm 0.006$ | $0.801 \pm 0.011$ | $0.760 \pm 0.016$ | $0.837 \pm 0.004$ | $0.774 \pm 0.044$ | $0.864 \pm 0.023$ |
| | AttnEigval | $0.794 \pm 0.055$ | $0.803 \pm 0.004$ | $0.779 \pm 0.006$ | $0.750 \pm 0.023$ | $0.833 \pm 0.006$ | $0.770 \pm 0.025$ | $0.864 \pm 0.017$ |
| | LapEigval | $0.775 \pm 0.047$ | $0.822 \pm 0.006$ | $0.820 \pm 0.020$ | $0.767 \pm 0.017$ | $0.859 \pm 0.003$ | $0.755 \pm 0.046$ | $0.862 \pm 0.025$ |
| | LookbackLens | $0.843 \pm 0.028$ | $0.828 \pm 0.004$ | $0.816 \pm 0.015$ | $0.767 \pm 0.020$ | $0.850 \pm 0.006$ | $0.773 \pm 0.043$ | $0.877 \pm 0.021$ |
| | MTopDiv | $0.845 \pm 0.035$ | $0.835 \pm 0.005$ | $0.818 \pm 0.022$ | $\mathbf{0.780 \pm 0.022}$ | $0.866 \pm 0.006$ | $\mathbf{0.797 \pm 0.043}$ | $0.872 \pm 0.019$ |
| | SinkProbe | $\mathbf{0.854 \pm 0.021}$ | $\mathbf{0.846 \pm 0.004}$ | $\mathbf{0.821 \pm 0.011}$ | $0.764 \pm 0.019$ | $\mathbf{0.877 \pm 0.006}$ | $0.761 \pm 0.046$ | $\mathbf{0.896 \pm 0.015}$ |
| | Δ | +1.1% | +1.3% | +0.1% | -2.1% | +1.3% | -4.5% | +2.2% |
| Llama3.1-8B | AttnScore | $0.752 \pm 0.048$ | $0.770 \pm 0.012$ | $0.677 \pm 0.017$ | $0.656 \pm 0.019$ | $0.678 \pm 0.011$ | $0.704 \pm 0.026$ | $0.722 \pm 0.026$ |
| | AttnLogDet | $0.812 \pm 0.034$ | $0.849 \pm 0.015$ | $0.759 \pm 0.018$ | $0.752 \pm 0.022$ | $0.827 \pm 0.014$ | $0.765 \pm 0.041$ | $0.825 \pm 0.024$ |
| | AttnEigval | $0.797 \pm 0.037$ | $0.850 \pm 0.011$ | $0.759 \pm 0.029$ | $0.761 \pm 0.012$ | $0.835 \pm 0.008$ | $0.773 \pm 0.042$ | $0.818 \pm 0.021$ |
| | LapEigval | $\mathbf{0.826 \pm 0.029}$ | $0.878 \pm 0.009$ | $0.787 \pm 0.027$ | $0.785 \pm 0.024$ | $0.874 \pm 0.013$ | $0.757 \pm 0.068$ | $0.834 \pm 0.016$ |
| | LookbackLens | $0.816 \pm 0.042$ | $0.879 \pm 0.007$ | $0.776 \pm 0.024$ | $0.776 \pm 0.019$ | $0.868 \pm 0.012$ | $0.767 \pm 0.040$ | $0.838 \pm 0.017$ |
| | MTopDiv | $0.803 \pm 0.036$ | $0.881 \pm 0.005$ | $0.772 \pm 0.021$ | $0.787 \pm 0.018$ | $0.874 \pm 0.009$ | $0.775 \pm 0.047$ | $0.809 \pm 0.014$ |
| | SinkProbe | $0.824 \pm 0.025$ | $\mathbf{0.890 \pm 0.005}$ | $\mathbf{0.789 \pm 0.026}$ | $\mathbf{0.798 \pm 0.020}$ | $\mathbf{0.883 \pm 0.012}$ | $\mathbf{0.778 \pm 0.042}$ | $\mathbf{0.879 \pm 0.009}$ |
| | Δ | -0.2% | +1.0% | +0.3% | +1.4% | +1.0% | +0.4% | +4.9% |
| Mistral-Nemo | AttnScore | $0.712 \pm 0.058$ | $0.688 \pm 0.013$ | $0.666 \pm 0.016$ | $0.641 \pm 0.019$ | $0.659 \pm 0.018$ | $0.678 \pm 0.053$ | $0.773 \pm 0.026$ |
| | AttnLogDet | $0.797 \pm 0.063$ | $0.797 \pm 0.006$ | $0.750 \pm 0.019$ | $0.749 \pm 0.013$ | $0.810 \pm 0.013$ | $0.798 \pm 0.048$ | $0.847 \pm 0.018$ |
| | AttnEigval | $0.780 \pm 0.062$ | $0.791 \pm 0.008$ | $0.730 \pm 0.026$ | $0.745 \pm 0.015$ | $0.815 \pm 0.014$ | $0.779 \pm 0.056$ | $0.826 \pm 0.017$ |
| | LapEigval | $0.804 \pm 0.049$ | $0.831 \pm 0.005$ | $0.765 \pm 0.028$ | $0.777 \pm 0.019$ | $0.860 \pm 0.003$ | $0.806 \pm 0.042$ | $0.847 \pm 0.007$ |
| | LookbackLens | $\mathbf{0.841 \pm 0.063}$ | $0.837 \pm 0.009$ | $0.782 \pm 0.022$ | $0.783 \pm 0.014$ | $0.866 \pm 0.009$ | $0.804 \pm 0.039$ | $0.866 \pm 0.017$ |
| | MTopDiv | $0.813 \pm 0.062$ | $0.837 \pm 0.009$ | $0.776 \pm 0.025$ | $0.770 \pm 0.012$ | $0.865 \pm 0.007$ | $0.811 \pm 0.030$ | $0.841 \pm 0.005$ |
| | SinkProbe | $0.811 \pm 0.036$ | $\mathbf{0.849 \pm 0.008}$ | $\mathbf{0.787 \pm 0.026}$ | $\mathbf{0.790 \pm 0.011}$ | $\mathbf{0.878 \pm 0.008}$ | $\mathbf{0.821 \pm 0.052}$ | $\mathbf{0.876 \pm 0.011}$ |
| | Δ | -3.6% | +1.4% | +0.6% | +0.9% | +1.4% | +1.2% | +1.2% |

**Results.** We report results for SinkProbe and the compared baselines in Table 1, where values denote the mean and standard deviation of ROC–AUC over 5-fold cross-validation. Across all evaluated datasets and model families, SinkProbe achieves the best performance in 23 of 28 model–dataset pairs, indicating that sink score–based features provide a robust signal for hallucination detection. The method performs consistently well across diverse tasks and domains (question answering and mathematical reasoning) and across models of varying scale (from 3B up to 12B parameters). While, LookbackLens, MTopDiv, and LapEigval often achieve competitive performance, all three are conceptually related to SinkProbe, particularly LapEigval.

## 4 CONCLUSION

We introduce SINKSCOREPROBE, a simple yet potent method for hallucination detection based on attention sink scores. Despite its simplicity, the approach offers far-reaching implications for understanding how models internalize hallucinations. We find the predictive signal is localized to specific heads and layers, yielding an interpretable internal signature. Furthermore, these sinks correlate with systematic differences in value norms, linking sink behavior to broader information flow in LLMs. Results across multiple benchmarks show that SINKSCOREPROBE achieves state-of-the-art performance compared to prior attention-based approaches.

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

## A  APPENDIX

### A.1  SINK SCORE AS AN UNDERLYING CONCEPT FOR HALLUCINATION DETECTION METHODS

In this section, we show that several existing attention-based hallucination detection methods can be interpreted through the lens of attention sink behavior. Although the concepts underlying these methods were originally developed from different motivations and formulated using distinct mathematical

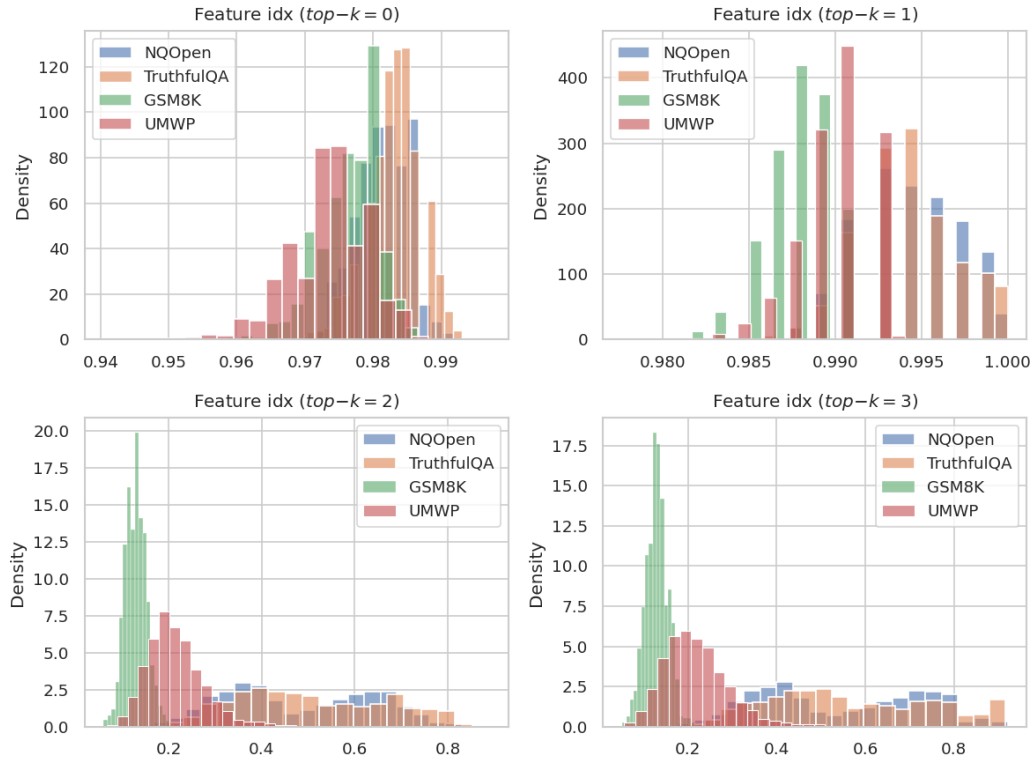

Figure 2: Distribution of the mean frequency with which the token with sink score at rank $k$ lies in the prompt (averaged over heads), for Llama3.2-3B.

constructions, we demonstrate that they implicitly rely on the concentration of attention onto sink tokens. This perspective provides a unifying explanation for their effectiveness and clarifies the role of attention collapse as a common underlying signal.

**LLMCheck (36)** was one of the first methods to leverage attention maps for hallucination detection, proposing AttentionScore, which aggregates per-layer self-attention terms into a scalar hallucination indicator. Specifically, for a given layer $l$, the score is defined as an average log-determinant of its attention maps:

$$\text{AttentionScore}^{(l)} = \frac{1}{HT} \sum_{h=1}^{H} \sum_{i=1}^{T} \log(\mathbf{A}_{ii}^{(l,h)}) \tag{1}$$

If a token has large sink scores, i.e., later tokens attend to it heavily, then their attention mass in concentrated outside the diagonal $\mathbf{A}_{ii}^{(l,h)}$, thus decreasing the log-determinant $\frac{1}{T} \sum_{i=1}^{T} \log(\mathbf{A}_{ii}^{(l,h)})$. Therefore, the tokens absorbing a substantial fraction of the attention mass disproportionately affect this score relative to non-sink tokens.

**LookbackLens (10)** proposed to detect contextual hallucinations by comparing attention allocated to the prompt versus attention allocated to the already generated response. The key assumption is that hallucinated generations are less grounded in the prompt, particularly in retrieval-augmented generation (RAG) settings. The central quantity is the lookback ratio. Let $|P|$ be the number of prompt tokens and $|R|$ be the number of response tokens. For each generated token $i$, define:

$$Attn_{ctx}^{(l,h)}(i) = \frac{1}{|P|} \sum_{u=1}^{|P|} \mathbf{A}_{iu}^{(l,h)}$$

$$Attn_{resp}^{(l,h)}(i) = \frac{1}{|R|} \sum_{u=|P|}^{|P|+|R|} \mathbf{A}_{iu}^{(l,h)}$$

Then, lookback ratio $LR_i^{(l,h)}$ for a layer $l$ and head $h$ is defined as:

$$LR_i^{(l,h)} = \frac{Attn_{resp}^{(l,h)}(i)}{Attn_{ctx}^{(l,h)}(i) + Attn_{resp}^{(l,h)}(i)} \qquad (2)$$

Attention sink scores capture complementary, column-wise structure by identifying tokens that consistently attract large amounts of attention across decoding steps. Although sink scores and lookback ratios are not algebraically related, they are coupled through the row-stochastic normalization of attention. When attention concentrates on a small set of sink tokens at a given step, the remaining tokens necessarily receive less attention.

This coupling has different implications depending on the location of the sinks. Sinks among generated tokens promote self-referential attention, increasing $Attn_{resp}^{(l,h)}$ and the lookback ratio, and are therefore closely associated with hallucinated outputs. In contrast, sinks located in the prompt concentrate attention on a narrow subset of prompt tokens without ensuring effective use of the full context. Consequently, prompt sinks reflect attention collapse rather than reliable grounding and do not consistently correspond to reduced hallucination risk. This distinction clarifies why lookback-based detectors are primarily sensitive to generated-token attention collapse, while sink-based diagnostics capture a broader class of attention concentration phenomena.

Subsequently, we analyze which parts of the input (prompt vs. response) fall within our proposed top-$k$ sink scores. Figure 2 shows that the highest-ranked sink ($k{=}0$) lies almost exclusively in the prompt across all datasets (mass near 1), consistent with it typically corresponding to $\langle bos \rangle$. The next sink ($k{=}1$) also falls predominantly in the prompt, with modest dataset-dependent variability. In contrast, lower-ranked sinks ($k{\geq}2$) are increasingly likely to occur in the generated answer; this shift is particularly pronounced on GSM8K (and to a lesser extent UMWP), whereas NQOpen and TruthfulQA retain a substantial prompt contribution. This finding further highlights that later sinks, which are crucial for hallucination detection, often originate from the generated answer rather than purely from the prompt (see Section D.2 for a study of the effect of $k$).

**Topological Divergence (7)** introduced the TOHA method, which, similarly to LookbackLens, relies on the assumption that there is a relationship between prompt and response tokens (the response should be grounded in the prompt) and was designed specifically for RAG scenarios.

TOHA formulates the attention map as a weighted graph with edge weights representing a pseudo-distance between tokens, e.g., $d_{ij} = 1 - a_{ij}$, where $a_{ij}$ denotes the attention weight between token $j$ and token $i$. Similarly to LookbackLens, tokens are partitioned into prompt tokens $P$ and response tokens $R$. The method computes probe features based on topological divergence, which in this setting can be reduced to computing the minimum spanning tree (MST) between a collapsed prompt node and the response tokens:

$$MTopDiv(P, R) = \sum_{(i,j) \in MST(P,R)} d_{ij} \qquad (3)$$

Since sink nodes can induce low pseudo-distances, they are likely to appear in the MST and thus can play a significant role in detection, as also observed in the original work.

**Spectral features (8)** proposed leveraging attention graphs and order statistics of graph Laplacian eigenvalues to detect hallucinations, motivated by the observation that the Laplacian can capture disruptions to information flow within the LLM. Surprisingly, we find that these eigenvalues correspond to sink scores discounted by self-attention. Due to the lower triangularity of both the attention and Laplacian matrices, the eigenvalues lie on the main diagonal entries:

$$l_{ii}^{(l,h)} = d_{ii}^{(l,h)} - a_{ii}^{(l,h)} = \frac{\sum_{u=1}^{T} a_{ui}^{(l,h)}}{T - i} - a_{ii}^{(l,h)}$$

By noticing the direct correspondence of the first term to the sink score, we can rewrite the eigenvalues as:

$$l_{ii}^{(l,h)} = s_i^{(l,h)} - a_{ii}^{(l,h)}$$

This relationship reveals that Laplacian-based features inherently encode sink score information while additionally discounting self-attention, thereby explaining their empirical effectiveness.

## B  RELATED WORK

**Information flow in LLMs**   Recently, there has been a surge of research analyzing the intricacies of LLMs from the perspective of information flow. In particular, interpreting attention maps as learned adjacency operators establishes a natural bridge between Transformers (38) and Graph Neural Networks (18): both propagate information through iterative mixing governed by an underlying graph structure (4). (6; 5) showed that over-squashing – a bottleneck phenomenon extensively studied in GNNs – can either impair information propagation or prevent excessive over-mixing. Beyond theoretical insights, this graph-theoretic framing opens the door to applying spectral, topological, and message-passing tools to analyze model internals (8; 7; 15). In this work, we focus on one dimension of information flow in LLMs, as quantified by the sink score metric, to detect hallucinations.

**Hallucination Detection via Attention Signals**   The interpretability afforded by attention-based analysis has motivated a line of work on detecting hallucinations in Transformer Decoder models (38) – fluent outputs unsupported by input or factual knowledge (3). (36) propose an attention score aggregating diagonal self-attention, observing lower values for hallucinated generations. (10) introduce the LookbackLens, contrasting attention to prompt versus response tokens to detect contextual hallucinations. Graph-based approaches have also proven effective: spectral features from Laplacian eigenvalues (8), topological features (7; 34), and neural message-passing on attention graphs (15) each capture complementary aspects of attention structure. Although these methods were developed from distinct motivations, we show in this work that some of them share a common dependence on attention sink behavior – a unifying perspective that motivates our sink-score-based approach and clarifies the mechanistic basis of attention-based detection.

**Attention Sinks**   A striking regularity observed across Transformer architectures is the emergence of *attention sinks* – tokens that attract disproportionate attention mass from subsequent positions regardless of semantic relevance. First identified in the context of efficient streaming inference (40), attention sinks have since been shown to emerge universally during pretraining (17). Rather than a pathology, recent work suggests sinks serve a functional role: (5) argue they prevent over-mixing by providing a stable routing target, while (33) offer a geometric characterization of sink formation. (30) unify attention sinks with the related phenomenon of compression valleys, showing both arise from massive activations in middle layers and reflect a transition from broad mixing to compressed computation. In this work, we show that commonly used metric for measuring a "sinkness", i.e., sink score, can provide rich signal to detect hallucinations.

## C  METHODOLOGY AND IMPLEMENTATION DETAILS

In our empirical study, we closely follow the methodology of (8), with three key modifications: (1) incorporating the UMWP dataset (41) to extend coverage to reasoning tasks, (2) employing cross-validation to improve reliability, and (3) removing the PCA component to enhance interpretability and facilitate analysis of the trained model. Our conclusions remain consistent regardless of whether PCA is applied.

For logistic regression, we used the scikit-learn implementation (29) with `max_iter=1000` and `class_weight=balanced`, leaving all other parameters at their default values. LLM inference was performed using the Transformers library (39) on NVIDIA A40 GPUs and A100 GPUs (48GB and 80GB VRAM, respectively). Our implementation, including model and prompt configurations, is available in the supplementary materials.

We perform inference on each dataset and recorded both the generated answers and the model's internal states, namely attention maps. We then employ `gpt-4.1` in an LLM-as-judge setting to evaluate model responses against gold answers for all datasets except GSM8K, where answers were

Table 2: Number of non-zero coefficients in the Logistic Regression model trained with $L_1$ regularization, i.e., number of sink score features selected for hallucination detection.

| LLM | Dataset | Total Features | Total Important |
|---|---|---|---|
| Llama3.2-3B | GSM8K | 6720 | 38 (1%) |
| | NQOpen | 6720 | 204 (3%) |
| | TruthfulQA | 6720 | 58 (1%) |
| | UMWP | 6720 | 228 (3%) |
| Mistral-Nemo | GSM8K | 12800 | 82 (1%) |
| | NQOpen | 12800 | 510 (4%) |
| | TruthfulQA | 12800 | 110 (1%) |
| | UMWP | 12800 | 177 (1%) |

verified programmatically. To ensure high-quality hallucination labels, we manually reviewed a random sample of the judge's evaluations. We exclude examples that exceeded the maximum token limit (2,048 for reasoning tasks; the maximum answer length for QA datasets) or lacked a verifiable answer. Dataset statistics are provided in Section E.2. Subsequently, we extract features and use them to train a logistic regression hallucination probe, reporting results for the optimal value of the $k$ hyperparameter. We employ 5-fold cross-validation, ensuring identical fold partitions across all methods for fair comparison.

# D ANALYSIS

To provide deeper insight into SinkProbe, this section analyzes which features drive hallucination detection. Throughout our experiments, we use models of varying sizes from the Mistral and Llama families (Mistral-Nemo and Llama-3.2-3B), evaluated on four datasets: NQOpen and TruthfulQA (question answering), and GSM8K and UMWP (mathematical reasoning).

## D.1 SINK-FEATURE IMPORTANCE IN HALLUCINATION DETECTION

Recall that hallucination detection is formulated as a classification problem over a feature vector derived from attention sink scores. For each attention head, we retain the top-$k$ largest sink scores and concatenate them across all heads and layers to obtain a fixed-dimensional representation, which is then fed to a logistic regression probe. To determine the importance of each sink score, we examine the probe's learned coefficients. Here, we train the probe with $L_1$ regularization to encourage sparsity in the coefficients while maintaining comparable performance to the unregularized setting[1]. Let $\beta_i^{(l,h)}$ denote the coefficient associated with the $i$-th largest sink score in layer $l$ and head $h$. For each model–dataset pair, we count the number of non-zero coefficients and report the results in Table 2. Notably, the probe retains only a small subset of features—between 1% and 4% of all sink-score features—corresponding to a few dozen to a few hundred coefficients.

Next, we study how sink-score importance is distributed across network depth. For each layer $l$, we define aggregate attention sink importance by summing the absolute values of coefficients associated with that layer,

$$I_l = \sum_h \sum_i \left| \beta_i^{(l,h)} \right|.$$

We visualize the resulting distribution in Figure 3. We find that informative aggregate importance sink scores are distributed across multiple layers, with the middle and final layers contributing most strongly. In light of recent work proposing a mix–compress–refine paradigm of LLM computation (30), this pattern suggests that heightened information mixing in intermediate and final

---

[1]To corroborate these findings with an alternative interpretability method, we performed a similar analysis using SHAP (25). While SHAP distributes importance across a broader set of features, the highest-ranked features largely overlap with those identified by the $L_1$-regularized model.

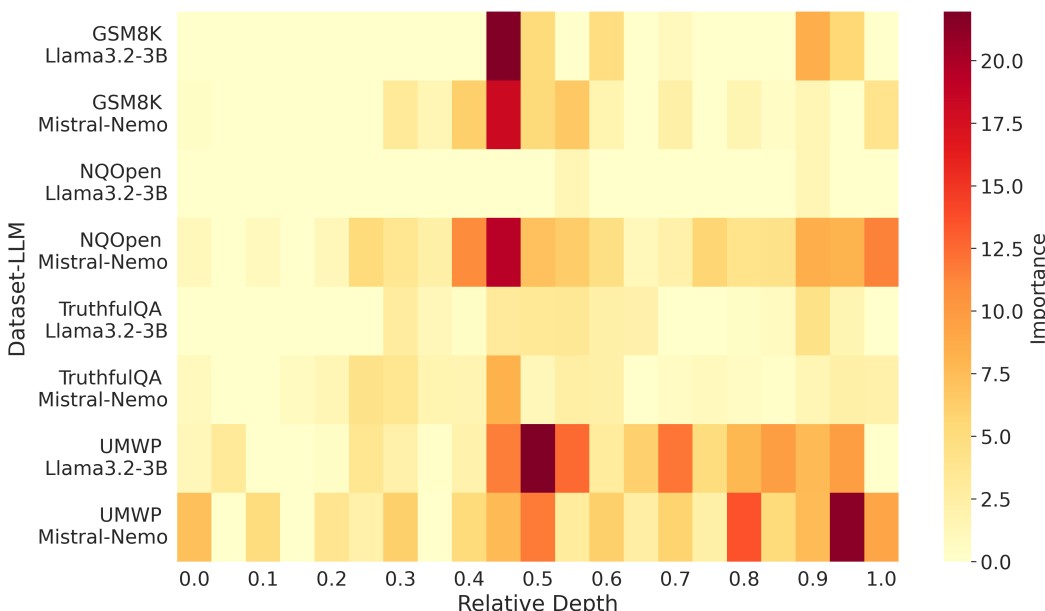

Figure 3: Importance scores of attention sinks derived from an $L_1$-regularized probe, aggregated across heads and sink indices per layer: $I_l = \sum_h \sum_i |\beta_i^{(l,h)}|$. Scores are plotted against relative layer depth to facilitate cross-model comparison.

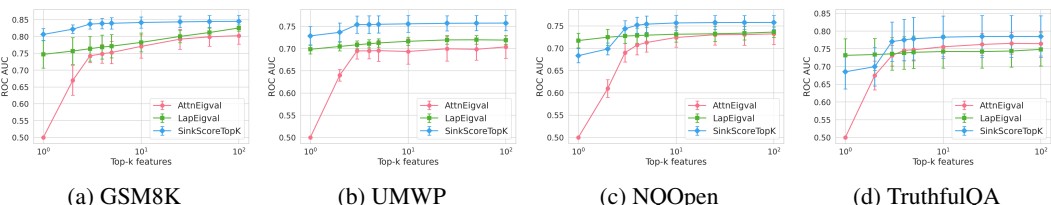

|            (a) GSM8K            |            (b) UMWP            |            (c) NQOpen            |            (d) TruthfulQA            |

Figure 4: Influence of $k$. Varying the number of retained sink scores per head, we find that SinkProbe attains best or near-best performance even for small values of $k$. This suggests that hallucination-related signals are concentrated in a few dominant sinks and that sink-score features capture this signal more directly and robustly than existing attention- or spectral-based representations. The presented results are for Llama3.2-3B, for other models and datasets see Section G.

layers may be a key driver of hallucination-related signals. This over-mixing could be caused by the atypically large value norms, which we examined in Section 2.1.

### D.2 HOW DOES TOP-$k$ AFFECT HALLUCINATION DETECTION?

In this study, we compare SinkProbe with other probing approaches that rely on top-$k$ feature selection, namely AttnEigvals and LapEigval. Figure 4 shows how hallucination detection performance varies with increasing $k$. We observe that both LapEigval and SinkProbe achieve their best or near-best performance for small values of $k$ ($k < 10$), whereas raw AttnEigvals improves more gradually with increasing $k$ but does not surpass either method. Notably, SinkProbe exhibits a pronounced performance gain for $k$ between 2 and 5, indicating that hallucination-related information is highly concentrated in a small number of dominant sinks.

This behavior highlights a key advantage of sink-score features over prior attention- and spectral-based representations. While methods such as AttnEigvals and LapEigval rely on more global aggregation of attention structure, sink scores isolate the most extreme attention attractors, yielding a more direct and robust signal. In particular, the leading sink often corresponds to the ⟨bos⟩ token (see Figure 2), which already provides a strong hallucination signal, further strengthened by incorporating

a small number of additional sinks. Consistent with prior work highlighting the role of attention sinks in information flow (5; 30), our results suggest that anomalies in this flow—manifested as disproportionate attention to non-informative tokens—are closely associated with hallucinated outputs. This provides a mechanistic interpretation of why sink-score features capture hallucination-related behavior more effectively than alternative attention-based descriptors.

### D.3  EXPERIMENTAL SETUP

**Datasets.**  We evaluate our method on seven diverse hallucination detection benchmarks spanning question answering and mathematical reasoning: *GSM8K* (11) contains grade school math word problems requiring multi-step reasoning; *UMWP* (41) provides university-level math word problems designed to probe model knowledge boundaries; *TruthfulQA* (24) tests factual accuracy on questions where models commonly produce misconceptions; *TriviaQA* (21) and *NQ-Open* (22) evaluate open-domain factual knowledge; *SQuADv2* (31) includes reading comprehension questions; and *HaluEvalQA* (23) provides a targeted hallucination evaluation benchmark. This varied selection ensures our evaluation covers both closed-book knowledge retrieval and context-grounded reasoning scenarios. Further details on the datasets and their preprocessing are provided in Section E.1.

**LLMs.**  We evaluate our method on four widely used open-weight LLMs spanning three model families and ranging from 3B to 12B parameters: Llama-3.2-3B (2), Phi-3.5 (4B) (1), Llama-3.1-8B (2), and Mistral-Nemo (12B) (27). Further details about the models are provided in Section F.

**Methodology**  We perform inference on each dataset and, recording generated answers and attention maps. Model responses are evaluated against gold answers using an LLM-as-judge setting, with selective manual review to ensure label quality. Then, we extract features for SinkProbe and baselines, and train a logistic regression hallucination probe, reporting ROC-AUC metric from 5-fold cross-validation. Methodology and implementation details are provided in Section C.

**Baselines.**  We compare SinkProbe against several attention-based hallucination detection baselines: (1) AttentionScore (36), an unsupervised method that computes the log-determinant of attention matrices aggregated across layers and heads, with AUC measured directly on raw scores; (2) AttnLogDet, a supervised counterpart of AttentionScore that uses attention log-determinants as features for a trained hallucination probe; (3) AttnEigvals and (4) LapEigval (8), which extracts the top-$k$ eigenvalues of attention and laplacian matrices, respectively; (5) LookbackLens (10), which measures the ratio of attention attributed to context; and (6) MTopDiv (7), which leverages topological features of attention graphs. All baselines except the unsupervised AttentionScore use the same logistic regression probe architecture and training protocol to ensure fair comparison.

## E  DATASET DETAILS

### E.1  UPSTREAM DATASET DETAILS

Here, we provide details on QA and reasoning datasets used to generate hallucination datasets. We selected them based on their prevalence in the literature regarding hallucination in LLMs. Further details are provided in Table 3.

Table 3: Detailed references for the datasets used in the experiments. ([†]Preprocessed following (9)).

| Dataset | Split / Subset | # Examples | Source |
|---|---|---:|---|
| NQ-Open (22) | Validation | 3,610 | huggingface |
| TriviaQA[†] (21) | Validation | 7,983 | huggingface |
| SQuADv2[†] (31) | Dev (`rc.nocontext`) | 9,960 | official website |
| HaluEvalQA (23) | QA | 10,000 | official repository |
| TruthfulQA (24) | Generation | 817 | huggingface |
| GSM8k (11) | Test | 1,319 | huggingface |
| UMWP (41) | *(entire dataset)* | 5,200 | official repository |

## E.2 DETAILS OF HALLUCINATION DATASET GENERATION

We followed the experimental pipeline from the main text to construct hallucination labels for all benchmarks. For each dataset, we ran model inference and stored the generated answers together with attention maps. We then used `gpt-4.1` as an LLM-as-judge to compare each answer against the gold reference and assign a hallucination label. For GSM8K, correctness was verified programmatically rather than with the judge. We manually inspected a random subset of judge decisions to ensure label quality. We filtered out responses that exceeded the 2,048-token limit or were deemed unverifiable by the judge (e.g., missing an explicit answer). The remaining examples form the dataset used for training and evaluation.

Table 4: Statistics of the obtained hallucination datasets. #Valid - number of verifiable, valid answers, #Invalid - number of invalid answers (answers which exceeded allowed number of tokens or did not contain answer (judge model marked them unverifiable)), #Non-Hallucinated - number of answers among Valid which were not hallucinated, #Hallucinated - number of answers among Valid which were hallucinated, Hallucination Ratio - ratio of hallucinated answers to the total number of valid answers.

| Dataset | LLM | #Invalid | #Valid Total | Valid Ratio | #Non-Hallucinated | #Hallucinated | Hallucination Ratio |
|---|---|---|---|---|---|---|---|
| TriviaQA | Llama3.1-8B | 368 | 9,592 | 0.96 | 6,423 | 3,169 | 0.33 |
| TriviaQA | Llama3.2-3B | 343 | 9,617 | 0.97 | 5,092 | 4,525 | 0.47 |
| TriviaQA | Mistral-Nemo | 31 | 9,929 | 1.00 | 6,599 | 3,330 | 0.34 |
| TriviaQA | Phi3.5 | 56 | 9,904 | 0.99 | 5,294 | 4,610 | 0.47 |
| NQOpen | Llama3.1-8B | 731 | 2,879 | 0.80 | 1,056 | 1,823 | 0.63 |
| NQOpen | Llama3.2-3B | 888 | 2,722 | 0.75 | 1,173 | 1,549 | 0.57 |
| NQOpen | Mistral-Nemo | 7 | 3,603 | 1.00 | 1,103 | 2,500 | 0.69 |
| NQOpen | Phi3.5 | 63 | 3,547 | 0.98 | 901 | 2,646 | 0.75 |
| SQuADv2 | Llama3.1-8B | 1,374 | 4,554 | 0.77 | 1,081 | 3,473 | 0.76 |
| SQuADv2 | Llama3.2-3B | 916 | 5,012 | 0.85 | 898 | 4,114 | 0.82 |
| SQuADv2 | Mistral-Nemo | 20 | 5,908 | 1.00 | 1,327 | 4,581 | 0.78 |
| SQuADv2 | Phi3.5 | 173 | 5,755 | 0.97 | 1,388 | 4,367 | 0.76 |
| TruthfulQA | Llama3.1-8B | 94 | 723 | 0.88 | 234 | 489 | 0.68 |
| TruthfulQA | Llama3.2-3B | 54 | 763 | 0.93 | 200 | 563 | 0.74 |
| TruthfulQA | Mistral-Nemo | 4 | 813 | 1.00 | 207 | 606 | 0.75 |
| TruthfulQA | Phi3.5 | 12 | 805 | 0.99 | 243 | 562 | 0.70 |
| HaluevalQA | Llama3.1-8B | 2,468 | 7,532 | 0.75 | 2,551 | 4,981 | 0.66 |
| HaluevalQA | Llama3.2-3B | 2,269 | 7,731 | 0.77 | 2,021 | 5,710 | 0.74 |
| HaluevalQA | Mistral-Nemo | 34 | 9,966 | 1.00 | 3,492 | 6,474 | 0.65 |
| HaluevalQA | Phi3.5 | 209 | 9,791 | 0.98 | 2,788 | 7,003 | 0.72 |
| UMWP | Llama3.2-3B | 5 | 5,195 | 1.00 | 4,021 | 1,174 | 0.23 |
| UMWP | Llama3.1-8B | 55 | 5,145 | 0.99 | 4,348 | 797 | 0.15 |
| UMWP | Mistral-Nemo | 6 | 5,194 | 1.00 | 4,399 | 795 | 0.15 |
| UMWP | Phi3.5 | 182 | 5,018 | 0.96 | 4,111 | 907 | 0.18 |
| GSM8K | Llama3.2-3B | 17 | 1,302 | 0.99 | 973 | 329 | 0.25 |
| GSM8K | Llama3.1-8B | 22 | 1,297 | 0.98 | 1,104 | 193 | 0.15 |
| GSM8K | Mistral-Nemo | 108 | 1,211 | 0.92 | 1,058 | 153 | 0.13 |
| GSM8K | Phi3.5 | 162 | 1,157 | 0.88 | 1,019 | 138 | 0.12 |

Table 4 summarizes the resulting sizes and label distributions across datasets and models. From a model-size perspective, invalid counts remain generally low but tend to be slightly higher for smaller models, which more often exceed length limits or omit explicit answers. Hallucination ratios also tend to decrease with scale: larger models are more consistently non-hallucinated across datasets, while smaller models show higher rates. This pattern aligns with improved instruction following and answer grounding as model capacity increases.

## F LLM INFERENCE DETAILS

In order to build our hallucination datasets, we leveraged 4 common LLMs from 3 different families, with sizes ranging from 3B up to 12B. We present specific versions of LLMs in Table 5. For QA datasets we adopted prompt from (28), while for GSM8K we used prompt from `lm-evaluation-harness` (16), and for UMWP we used prompt from (37). During inference, we set temperature to 0.1 and run all examples with batch size equal to 1 for reproducibility (19). Further details can be found in the supplementary material.

Table 5: LLM details.

| LLM | HuggingFace Repository |
|-----|------------------------|
| Llama3.2-3B (2) | meta-llama/Llama-3.2-3B-Instruct |
| Llama3.1-8B (2) | meta-llama/Llama-3.1-8B-Instruct |
| Mistral-Nemo (27) | mistralai/Mistral-Nemo-Instruct-2407 |
| Phi3.5 (1) | microsoft/Phi-3.5-mini-instruct |

# G  OPTIMAL $k$ HYPERPARAMETER VALUES

We run top-$k$ selection among several candidate values $j \in \{1, 2, 3, 4, 5, 10, 25, 50, 100\}$. In Table 6 we present which value of $k$ led to the best results. While we can observe that the best mean scores are achieved for the highest value of top-$k$, we notice that similar efficacy is obtained for small values of $k$, which can be observed in Figure 5.

Table 6: Values of top-$k$ providing best quality

| LLM | Dataset Feature | GSM8K | HaluevalQA | NQOpen | SQuADv2 | TriviaQA | TruthfulQA | UMWP |
|-----|-----------------|-------|-----------|--------|---------|----------|-----------|------|
| Llama3.2-3B | AttnEigval | 100 | 100 | 100 | 100 | 100 | 50 | 100 |
| | LapEigval | 100 | 100 | 100 | 100 | 100 | 100 | 50 |
| | SinkProbe | 100 | 50 | 100 | 25 | 100 | 50 | 100 |
| Phi3.5 | AttnEigval | 100 | 50 | 100 | 100 | 100 | 100 | 100 |
| | LapEigval | 10 | 50 | 100 | 100 | 100 | 100 | 100 |
| | SinkProbe | 50 | 50 | 50 | 100 | 50 | 2 | 50 |
| Llama3.1-8B | AttnEigval | 100 | 100 | 100 | 100 | 100 | 100 | 100 |
| | LapEigval | 100 | 100 | 100 | 100 | 100 | 10 | 100 |
| | SinkProbe | 3 | 100 | 100 | 25 | 100 | 100 | 100 |
| Mistral-Nemo | AttnEigval | 100 | 100 | 100 | 100 | 50 | 100 | 100 |
| | LapEigval | 100 | 25 | 50 | 100 | 25 | 100 | 100 |
| | SinkProbe | 2 | 50 | 100 | 100 | 25 | 100 | 100 |

# H  IMPORTANT HEADS ANALYSIS

To identify which attention heads are most predictive of hallucination, we employ $\ell_1$-regularized logistic regression on our sink score features. We fit an $\ell_1$-regularized logistic regression model. The $\ell_1$ penalty induces sparsity, setting most coefficients $\beta_i$ to exactly zero, thereby performing implicit feature selection. We set $C = 0.75$ based on 5-fold cross-validated ROC-AUC, balancing model sparsity with predictive performance.

**Important Head Selection.**  We identify important heads by examining the non-zero coefficients after $\ell_1$ regularization. For coefficient $\beta_{l,h,k}$ corresponding to layer $l$, head $h$, and top-$k$ position $k$, we compute the odds ratio $\exp(\beta_{l,h,k})$. Features with $|\exp(\beta_{l,h,k}) - 1| > \epsilon$ (where $\epsilon = 10^{-6}$) are retained as important predictors: positive coefficients indicate that higher sink scores increase hallucination probability, while negative coefficients indicate the opposite.

**Alternative: SHAP-based Selection.**  As an alternative to coefficient-based selection, we also considered SHAP values (25) for quantifying feature importance. For each feature, we compute the mean absolute SHAP value across training samples and select features exceeding a specified quantile threshold. Although SHAP provides model-agnostic importance scores that account for feature interactions, we found that $\ell_1$-regularized coefficients yield sparser selections, and that the highest-ranked SHAP features overlap with those identified by the logistic regression model.

**Layer-wise Importance Aggregation.**  To visualize importance distribution across model depth, we aggregate the absolute coefficient magnitudes by layer:

$$I_l = \sum_{h=1}^{H} \sum_{k=1}^{K} |\beta_{l,h,k}| \tag{4}$$

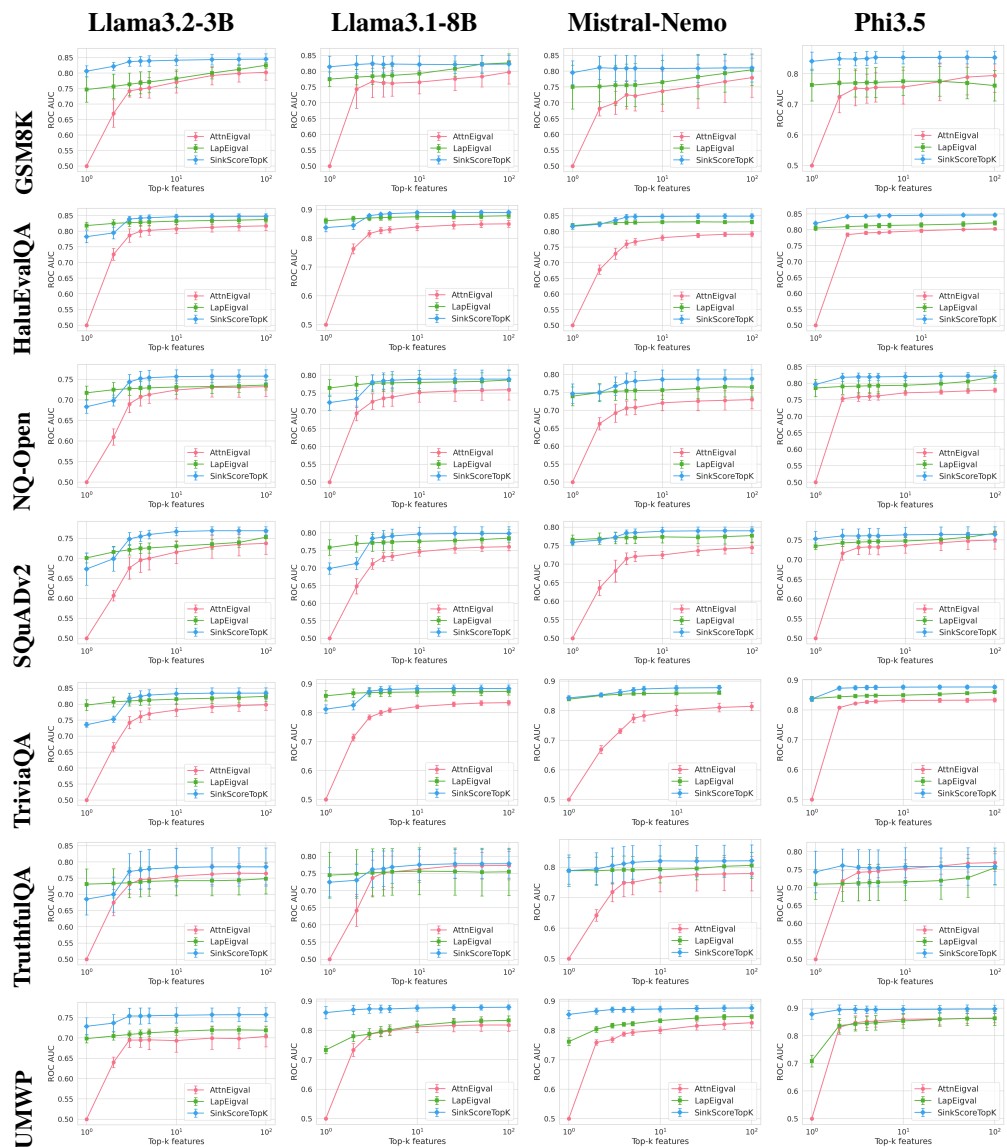

Figure 5: Hallucination detection performance (ROC-AUC) as a function of top-$k$ across all models and datasets. Each subplot shows the ROC-AUC performance on 5-fold cross-validation for different values of $k \in \{1, 2, 3, 4, 5, 10, 25, 50, 100\}$ and models. Columns represent different LLMs, rows represent different datasets.

In our visualizations, we normalize layer indices to relative depth $\delta_l = l/L$ to enable comparison across models with different numbers of layers. The relative depths are discretized into bins of width 0.05, and importance values are averaged within each bin to produce the heatmap visualization shown in Figure 3.

