# OpenReview forum: "Attention Sinks as Internal Signals for Hallucination Detection in Large Language Models"
_ICLR.cc/2026/Workshop/Sci4DL — Sci4DL 2026_

### Official Review · Reviewer_72dg · 2026-02-23

**Fit:** 3
**Significance:** 2
**Confidence:** 2

**Summary:**

This paper explores the connection between attention sinks, tokens that receive a significant amount of attention within attention layers relative to their semantic importance and hallucination. The paper begins by describing attention sinks and recalling the notion of a *sink score*. It then explains how the sink scores can be converted into a feature vector that can be used for downstream probes. Along the way, the paper analyzes in what cases sink scores may imply hallucination, focusing on the norm of the corresponding value vector. Finally, SinkProbe (the method proposed by the work) is compared against other similar attention-based hallucination detectors, showing intriguing performance.

**Strengths:**

- **Clarity:** The paper is well-written and easy to read. The main contributions are motivated and put into context in the introduction. The reviewer appreciated the reminder on sink score.
- **The observation and method are intuitive:** Given recent interest in attention sinks, it is natural to ask whether they play a role in hallucination detection. This paper offers evidence that these two phenomena are correlated. This reviewer would be interested in reading a longer, more in-depth study.

**Suggestions:**

- **Partition between main body and supplementary material:** The paper should be written so that the main contributions are at least partially covered in the main body of the text. One of the claims in the introduction is that “We identify computationally active sinks, showing that sink-based signals are strongest when attention concentration coincides with large value vector norms.” But to this reviewer’s memory, this is never actually performed in the main body.
- **The role of value norm:** The paper makes the interesting observation that sinks only seem to be associated with hallucination when the value norm is high. To the reviewer’s memory, this is never included as a feature in SinkProbe and is rather a post-hoc observation. Why not use this in the probe?
- **Why do attention sinks cause hallucination?:** Having pinpointed a possible explanation for some types of hallucination, it would be interesting to dig into the mechanics of how an attention sink leads to hallucination in some specific examples.

---

### Official Review · Reviewer_ZoZc · 2026-02-26

**Fit:** 2
**Significance:** 2
**Confidence:** 2

**Summary:**

The authors present a method to detect hallucinations in LLM's, a score based upon the target architecture itself, rather than at test time. This work has clear applications as a growing number of sectors depend on generative model outputs. The manuscript is well written and results presented clearly. The key findings in Table 1 show gains over alternative methods, and, though consistent, rarely exceed 2% ROC-AUC. The authors go on to claim that their methods are simple, which, I presume, means also computationally efficient, and this could be a key advance. However, they present no benchmark regarding execution time or similar which might better highlight the strengths of this method, given that table 1 metric improvements are modest.

**Strengths:**

- Clearly written with sufficient background
- Timely given growing societal dependency on LLM's
- Thorough factorial sweep of models and datasets
- Consistent, though modest, performance gains

**Suggestions:**

- The authors do not discuss limitations of this work
- Adoption of this technique over existing methods would be more compelling if the authors could demonstrate gains in efficiency

---

### Official Review · Reviewer_mQR6 · 2026-02-27

**Fit:** 1
**Significance:** 1
**Confidence:** 3

**Summary:**

This paper proposes SinkProbe, a novel hallucination detection method for Large Language Models (LLMs) based on attention sink scores—tokens that attract disproportionate attention mass during generation. The authors hypothesize that hallucinations are linked to breakdowns in internal information flow, transitioning from input-grounded attention to compressed, prior-dominated computation.

**Strengths:**

1. SinkProbe consistently outperforms six existing attention-based hallucination detection baselines, achieving the best results in 23 out of 28 tested model-dataset pairs across various LLM families.
2. The paper provides a principled mathematical framework that unifies several previous detection methods, demonstrating that spectral and graph-based approaches implicitly rely on attention sink behavior.

**Suggestions:**

Here are the weakness:
1. The predictive signal is heavily localized in the middle and final layers of the network, meaning the method may be less effective if applied to models with significantly different internal information flow dynamics.
2. While the method is robust, its performance is tied to the selection of the $k$ value (the number of top sinks retained); although small values of $k$ often work well, the optimal value can vary by dataset and model

---

### Meta-Review · Area_Chair_SsR1 · 2026-03-01

**Recommendation:** Accept

**Metareview:**

Based on the reviews, I recommend accept.

---

### Decision · Program_Chairs · 2026-03-02

Accept